# Class Explanations: the Role of Domain-Specific Content and Stop Words

**Denitsa Saynova[1], Bastiaan Bruinsma[1], Moa Johansson[1], Richard Johansson[1,2]**
[1]Chalmers University of Technology, Gothenburg, Sweden
[2]University of Gothenburg, Gothenburg, Sweden
{saynova, sebastianus.bruinsma, moa.johansson}@chalmers.se
richard.johansson@cse.gu.se

## Abstract

We address two understudied areas related to explainability for neural text models. First, *class explanations*. What features are descriptive across a class, rather than explaining single input instances? Second, the *type of features* that are used for providing explanations. Does the explanation involve the statistical pattern of word usage or the presence of domain-specific content words? Here, we present a method to extract both class explanations and strategies to differentiate between two types of explanations – domain-specific signals or statistical variations in frequencies of common words. We demonstrate our method using a case study in which we analyse transcripts of political debates in the Swedish Riksdag.

## 1 Introduction

Recent developments in NLP are often the result of ever more complex model architectures and an increasing number of model parameters. Yet, if we want to rely on these models, we should be able to review the similarities and dissimilarities between the model and human judgement. Explainability frameworks can do this by highlighting on *what* the model has learnt to base its decisions. Are these coincidental statistical patterns or something that a human would use as an explanation? Madsen et al. (2022) argue that explanations should ideally be both *functionally-grounded* (true to the underlying machine learning model) as well as *human-grounded* (useful to a human).

In this article, we propose a new method for extracting class explanations from text classifiers. Besides, we also show a new way to distinguish between two types of features that appear in those explanations, that is, between informative content words and subtle statistical differences in common words' frequencies. Our method aggregates explanations for individual data points (here provided by LIME (Ribeiro et al., 2016)), followed by a sorting stage that separates the different kinds of features.

Our work is in part motivated by use cases of machine learning for texts in the social sciences. In this field, explainability methods are relevant both as checks to compare against human expert knowledge and as a tool for bias detection. As a case study, we use our method to explain the decisions of a binary classifier trained to identify if speeches in the Swedish Riksdag belong to either of the two main parties, the Moderates (M) or the Social Democrats (S).

We find that our method can separate class explainability features and that those data points whose explanations contain primarily domain-specific content words are more often classified correctly.

## 2 Literature Review

As a result of the extensive work on explainability methods, a complex typology of different approaches exists (see Danilevsky et al. (2020) or Madsen et al. (2022) for a survey). One important distinction is between *global* and *local*. On the one hand, global methods aim to explain some general behaviour of a model, such as class explanations, which summarise the model with respect to a certain class. On the other, local methods aim to explain why the model assigned a single data point to a particular class.

Between global and local methods, the latter receive the most attention (Nauta et al., 2022). Three popular methods are gradient-based approaches (Baehrens et al., 2010), Shapley values (Shapley, 1952), and LIME. Gradient-based approaches use the model's weights and take the gradient with regard to the input. As such, they measure the

change in the outcome given some small change in the input. Yet, they are only an accurate reflection of the model if that model is linear (Li et al., 2016), which is not the case for most deep NLP architectures. On the other hand, while Shapley values have many theoretical guarantees to make them a faithful interpretation (they represent the true contributions of the features (Ethayarajh and Jurafsky, 2021)), their implementations (e.g. via attention flows for transformer-based architectures (Abnar and Zuidema, 2020)) tend to be computationally expensive, which is problematic in the current setting, where we focus on aggregating a substantial number of individual explanations. Finally, LIME has an advantage over gradient-based approaches as it is model agnostic. This means that LIME attempts to explain a trained classifier independently of its architecture (Ribeiro et al., 2016).

## 2.1 Class explanations

The area of global *class explanations* is so far less studied than that of local explanations. One approach to providing global understanding of the model is to use behavioural or structural probes (Tenney et al., 2019; Hewitt and Manning, 2019; Wallace et al., 2019). Probing is a technique where a supervised model (a probe) is used to determine what is encoded in the internal representation of the studied model. This is done by training the probe to predict based on the frozen representations of the black-box model. If the probe performs well on the task, that indicates the required information was well represented by the black-box model, if the probe is unable to achieve high accuracy, that is taken to signify that the studied patterns are not learned by the black-box model. This has some limitations – for example, the complexity of the probe. If the probe is too simple, it may not capture second order effects, if it is too complex, it may learn the task internally and "discover" things that are in the probe rather than the model (Hewitt and Liang, 2019). More importantly, these methods tend to be applied to the discovery of simple syntactic structures like part of speech (POS) tagging, syntactic tree structures (Rogers et al., 2020) or to detect the presence of specific knowledge (Petroni et al., 2019). Other attempts in this area include leveraging local methods and utilising a strategy for aggregating and presenting those results to the user. An example of such approach is SP-LIME (Ribeiro et al., 2016), which aggregates individual LIME explanations with a greedy search for finding data points (texts) that are explained by the most dissimilar sets of features in order to represent the breadth of the class explanations. The results are presented as ranked text examples with their corresponding explanations, where the number of examples is defined by the user. Due to its focus on features that cover as many input instances as possible, this method tends to overemphasise stop words (see further discussion in Section 6).

## 2.2 Features of Explanations

To a human, not all features learnt by the machine learning model are equally informative. Some signals may come from speech patterns, others from the topic that is discussed and the sentiment, yet others may indicate preferred catchphrases and slogans. There is a distinction between explanations of the model (what a model bases its prediction on) and human explanation (what a human would base their decision on if faced with the same prediction task) (Miller, 2019). Since humans have background knowledge that is not accessible to the model and the model has the capacity to detect small statistical signals that are beyond human computational capabilities, the set of features that are selected by either may differ. This issue can be viewed in terms of the concepts presented in the position paper by Doshi-Velez and Kim (2017) and further discussed by Madsen et al. (2022), namely – *human-grounded* and *functionally-grounded* explainability. Functionally-grounded explainability is concerned with how well the explanation reflects the model, whereas human-grounded explainability is concerned with producing explanations that are useful to a human. This is also in line with work by Nauta et al. (2022), where the authors argue for the rigorous evaluation of an explainability method across twelve properties in three categories – content, presentation, and user. The content properties and in particular *correctness* (faithfulness w.r.t. the black box) are related to the functionally-grounded approach, whereas the user properties – *context* (how relevant the explanation is to the user), *coherence* (how accordant the explanation is with prior knowledge), and *controllability* (how interactive or controllable an explanation is) – relate to human-grounded explainability.

In our work, we use stop words and content

words to align with functionally-grounded and human-grounded explanations. *Content words* are words that have independent meaning outside of the sentence they appear in. These are typically a noun, verb, adjective, or adverb and are distinguished from *function words*, which mainly express grammatical relationships and have little semantic content. *Stop words* are words that carry little or no important information for the task at hand and tend to contain *function words*. This concept is not strictly defined, but generally refers to high-frequency terms. It can therefore extend to, for example, procedural language (e.g. "tallman" (speaker)) that can also act as a stop word in the domain of Swedish political debates. A model can learn to detect distributional differences of any word as long as it is correlated with the predicted class, but a human will be unlikely to relate and understand the cause of the distributional differences of stop words. The difference in frequency of how often a group uses the word "also", for example, may not be very informative for a human, even if those distributional differences point to real speech patterns that distinguish between the speakers (Arun et al., 2009a) and have even been linked to the author's gender (Arun et al., 2009b). Human domain knowledge will most likely be captured through domain-specific, content words. Being able to confirm the (extent of the) model's grounding in content words can serve to validate it.

## 3 Method

Our algorithm for computing class explanations consists of four steps: post-hoc instance explanations extraction, aggregation, sorting, and a keyword-in-context search that extracts example texts. This framework is formalized in Algorithm 1. It is similar to SP-LIME, but rather than searching for data points that capture the most diversity of the important features, we propose to work directly with the feature importance and explore ways to summarize and sort these by relevance.

The replication materials and full results are available online [1].

### 3.1 Step 1: Instance explanation extraction

For a set of held-out data samples $N$, we apply the trained classifier $f$. In the instances where

---

**Algorithm 1** Class explainability from instance explanations

---

**Require:** Binary classifier $f$, data samples $N$
**Require:** Instance explainability function $g$
**Require:** Feature scoring function $h$
$\quad W \leftarrow \{\}$ $\qquad \triangleright$ features and importance scores
$\quad c1 \leftarrow \{\}$ $\qquad \triangleright$ features explaining class 1
$\quad c2 \leftarrow \{\}$ $\qquad \triangleright$ features explaining class 2

---

*Step 1 – Instance explanation extraction*

---

**for** $text, true\_label \in N$ **do**
$\quad$ **if** $f(text) = true\_label$ **then**
$\quad\quad$ $W \leftarrow W \cup \{g(text, f)\}$
$\quad$ **end if**
**end for**

---

*Step 2 – Aggregation*

---

**for** $feature, score \in W$ **do**
$\quad$ **if** $score < 0$ **then**
$\quad\quad$ $c1 \leftarrow c1 \cup \{feature\}$
$\quad$ **else**
$\quad\quad$ $c2 \leftarrow c2 \cup \{feature\}$
$\quad$ **end if**
**end for**

---

*Step 3 – Sorting*

---

**for** $c \in \{c1, c2\}$ **do**
$\quad$ return $c$ sorted by $h$ score
**end for**

---

*Step 4 – Keywords in context*

---

**for** $c \in \{c1, c2\}$ **do**
$\quad$ **for** $term \in$ top $X$ terms in $c$ **do**
$\quad\quad$ return all occurrences of $term$
$\quad\quad$ with $n$ words before and after
$\quad$ **end for**
**end for**

---

the classifier makes the correct prediction, we extract the list of features and their corresponding saliency with model $g$. This can also be flipped to focus on instances where the model makes the incorrect predictions to investigate which patterns or instances are hard to classify. A certainty threshold can also be used to explore only cases where the model is certain or borderline cases. Our method aims to be extendable to different model architectures, therefore we require a post-hoc, model agnostic instance explanation function $g$. For now, we have chosen LIME, but alternative

---

[1]`https://github.com/dsaynova/NoDaLiDa2023`

methods can be used as well, as long as they are able to extract features and the feature contribution scores that explain an instance. This means we are currently constrained by LIME's limitations and only consider single tokens as features. Since LIME is a surrogate model, there is also some uncoupling between the classification model and the explanations. For each correctly classified instance, we extract the top $k$ features (here set to 10). This can be reduced even further in order to limit the number of features that are considered or extended to include all tokens and the task of limiting the explanation will then be completely relegated to the sorting step.

## 3.2 Step 2: Aggregation

A feature can contribute either positively or negatively towards the prediction of the model. When working with a binary classifier, a negatively contributing feature towards predicting class 1 means it is a positively contributing feature for class 2. Therefore, the features collected from the previous step are aggregated in two sets – $c1$, $c2$ – one for each class based on their feature score sign. Note that these two sets of features may have overlaps if the predictive signal is indicative of the different context in which those features appear.

## 3.3 Step 3: Sorting

The resulting sets of features for each class need to be constrained to a feasible size to be interpretable by a human. We propose two approaches to developing a feature relevance score $h$ to prioritize and distinguish these terms along an axis of more domain-specific concepts to more generic words – *normalization* and *PCA*.

**Normalization.** Here, we use the sum of LIME scores for each feature of the explanation divided by number of occurrences of that feature in the validation set. We calculate the feature relevance score $h$ of the $j^{th}$ feature as: $h_j = \frac{1}{m_j} \sum_{i=1}^{N} W_{ij}$. Here, $N$ is the number of data points in the explained dataset, $m_j$ is the number of occurrences of feature $j$ in the explained set, and $W$ is the explanation matrix containing the local importance of the interpretable components for each instance. This will give higher scores to features identified as more important by LIME, but will penalise common words, if they do not contribute to a class prediction often. This is in line with the definition of stop words and should target the corpus-specific stop words. We also filter out words that appear in two or less documents, as these can be party specific, but may not be useful for generalisation. This number can also be increased to filter out more predictive (according to LIME) words.

**PCA.** The second approach to sorting is to decouple it from the LIME score after the initial aggregation step and use PCA of word embeddings. We found that PCA applied to pre-trained word embeddings tends to separate domain specific words from more generic terms. A theoretical motivation for this analysis lies in the distributional differences between a general text (used for pre-training word embeddings) and a domain-specific text (in this case – political debate). We hypothesise that the general embedding model will see the domain specific terms in sufficiently distinct context in order to embed them in a compact space with a latent dimension separating them from more common and general terms. This relies on the studied data having a significant amount of domain specific terminology that is rarer in general. We expect this to be the case for many application within the social sciences (e.g. politics), but can have limitations in, lower-level, syntactic classification tasks like POS tagging.

To calculate the sorting score, the terms from each set $c1$ and $c2$ are embedded using a model[2] trained on the Swedish CoNLL17 corpus. A PCA is run on each set of words – $c1$, $c2$ – and the first PCA dimension value is used as the sorting score $h$. Similarly to the normalisation approach, words that appear in two or fewer documents are filtered out. This dimension seems to provide a good distinction of domain specific terms.

## 3.4 Step 4: Keywords in Context

To further increase human interpretability, we also provide a way to provide context by extracting snippets of texts around the top word features produced in Step 3. For each occurrence, we use a simple keyword-in-context search and extract $n$ words before and after our feature word. This is clearly not feasible or interesting for very frequent words, which further motivates separating rarer, domain specific content words from more common stop words.

---

[2]http://vectors.nlpl.eu/repository/20/69.zip

## 4 Data

The dataset used for the case-study consists of transcripts of debates in the Swedish Riksdag, sourced from Riksdagens öppna data – Anföranden[3]. We use a pre-processed version available from Språkbanken[4] consisting of debates from 1993 to 2018. For our experiment, texts from the Social Democrat (S) and Moderate (M) parties have been extracted, resulting in 104,842 S and 62,160 M data points (one data point is one speech that could be part of a longer debate). From these, 100 examples have been sampled for a small-scale human baseline check, where two annotators are asked to perform the classification task of determining the party label from the speech texts and were evaluated against the true label. Since these are debates, references to the opponent are a strong but trivial predictor of party. References to people and political parties have been removed by targeting Swedish political party names' stems (for a full list please refer to the linked code base) and words tagged as "People_along_political_spectrum" in Språkbanken's tags, based on Swedish FrameNet (Heppin and Gronostaj, 2012). Since the cleanup is based on a coarse rule for party name stems detection and the automatic tags from Språkbanken, not all references have been removed. We have opted for blanking all certain cases, so that enough of the interfering signal is removed to make the classification task non-trivial, rather than applying a comprehensive and exhaustive search of all mentions, since that is not the main goal of this work. Data points shorter than 50 words have been removed, as manual analysis shows these tend to be entirely procedural and do not carry political sentiment. This is in line with similar cleaning practices used for US congressional debates (Bayram et al., 2019). The data is undersampled to balance the classes and split into: train (108,169), test (12,019) and validation (2,000) sets. The validation set is used for explainability methods.

## 5 Experiments

To test our methodology we apply it to a BERT classifier trained to predict the party label of a text (Devlin et al., 2019). The classifier is fine-tuned from a pre-trained model for Swedish data released by The National Library of Sweden/KBLab and available through the huggingface library[5]. The model has a 50,325 word vocabulary and 512 maximum token length. Longer inputs are truncated. As a baseline for investigating class differences and separability of the data we use a logistic regression classifier, as this provides easy access to class explanations by simply looking at the top and bottom scoring internal weights of the model. N-gram spans from 1 to 3 and a combination of all have been compared. The number of input features is 50,325 – the same as the pre-trained BERT model.

A small-scale human annotation check on 100 instances shows the two annotators perform with 58 and 56 percent accuracy respectively. A Cohen's kappa of 0.4 indicates this is a hard classification task.

In the interest of space, the sections below contain partial results. The full results are available online.

### 5.1 Baseline

Table 1 summarises the accuracy and F1 scores for the logistic regression classifier. We observe that the best result is achieved with 1-grams, with the inclusion of 2- and 3- grams adding no performance gains. It seems the main part of the distinguishing signal can be picked up by specific words rather than phrases.

| n-gram span | # feat | acc | F1 |
| --- | --- | --- | --- |
| 1,1 | 50,325 | 76.94 | 76.80 |
| 2,2 | 50,325 | 73.19 | 73.05 |
| 3,3 | 50,325 | 69.39 | 69.15 |
| 1,3 | 150,975 | 76.93 | 76.80 |

Table 1: Logistic regression classifier performance.

From the internal model weights, we can identify both domain specific words – "sjuka" (sick), "arbetslösa" (unemployed), "arbetslinjen" (the employment line, a Moderate catchphrase), and stop words – "det" (the), "också" (also), "synnerhet" (in particular), can be predictive of the party label. This is in agreement with our assumption that a model can depend on both statistical differences in stop word or in human concepts as

[3] https://data.riksdagen.se/data/anforanden/
[4] https://spraakbanken.gu.se/resurser/rd-anf-1993-2018
[5] https://huggingface.co/KB/bert-base-swedish-cased

the basis of its prediction, and in doing so outperforms the human annotators.

## 5.2 BERT

The BERT model (lr = 5e-6, batch size = 48, steps = 6000) shows only slight improvement over the baseline, summarised in Table 2.

| Evaluation | acc | F1 |
|---|---|---|
| test set | 78.44 | 76.66 |
| validation set | 79.95 | 78.27 |

Table 2: BERT classifier performance.

Applying LIME to all validation samples and aggregating the top 10 features for each data point results is a list of 2,043 Moderate and 2,085 Social Democrats terms. Out of these 1,456 Moderate and 1,334 Social Democrat terms appear in more than two documents, and are thus candidates to be included as part of class explanations (this limit can be adjusted by the user).

## 5.3 Validation

Tables 3 – 4 show the results of both LIME and PCA for both M and S. In both cases, the models separate informative terms from generic ones. This is especially the case with the LIME scores, where the lowest-scoring words are all stop words. As for the highest-scoring words, we find that they are all related to taxes and employment. This is understandable, as this is also what makes up the main political left/right dimension in Sweden (Franzmann and Kaiser, 2006; Jolly et al., 2022; Ezrow et al., 2011). Besides, we can identify several references to several (groups of) parties and ministers, which we would expect in debates. As discussed in section 3.2, we also find a term that appears as important for both parties - *budget-propositionen* (the budget bill). This is a result of the explainability model using single tokens as features and most likely indicates that this is a term mentioned in a different context for both parties.

While these findings are hopeful on their own, to be useful for social scientists, we need to do

| PCA ordering | |
|---|---|
| **rank** | **term** |
| 1 | utgiftsområde (expenditure area) |
| 2 | budgetpropositionen (the budget bill) |
| 3 | jobbskatteavdrag (employment tax credit) |
| 4 | arbetslöshetsförsäkringen (unemployment insurance) |
| 5 | skattehöjningar (tax increases) |
| | ... |
| 1454 | högkvalitativa (high quality) |
| 1455 | vackra (beautiful) |
| 1456 | klassiska (classic) |
| **Normalised LIME score** | |
| **rank** | **term** |
| 1 | vänsterregering (left-wing government) |
| 2 | fattigdomsbekämpning (poverty alleviation) |
| 3 | bidragsberoende (benefits dependency) |
| 4 | fridens (of peace) |
| 5 | arbetsföra (able to work) |
| | ... |
| 1454 | som (as) |
| 1455 | ett (one) |
| 1456 | en (one) |

Table 3: Results for the Moderates.

| PCA ordering | |
|---|---|
| **rank** | **term** |
| 1 | budgetpropositionen (the budget bill) |
| 2 | arbetsmarknadspolitik (labor market policy) |
| 3 | samlingspartiet [Refers to the Moderates] |
| 4 | ungdomsarbetslösheten (youth unemployment) |
| 5 | skattesänkningar (tax cuts) |
| | ... |
| 1332 | tillsammans (together) |
| 1333 | u (u) |
| 1334 | dam (lady) |
| **Normalised LIME score** | |
| **rank** | **term** |
| 1 | överläggningen (the deliberation) |
| 2 | moderatledda (moderate-led) |
| 3 | kd (abbrev. for Christian Democrat party) |
| 4 | skattesänkningarna (the tax cuts) |
| 5 | borgarna (the bourgeois [parties to the right]) |
| | . . . |
| 1332 | har (have) |
| 1333 | av (of) |
| 1334 | för (for) |

Table 4: Results for Social Democrats.

more to ensure that our results are *valid*. In other words, we want to ensure that our method measures what we intend to measure (Carmines and Zeller, 1979). In our case, this is whether a speech is representative of S or M.

Looking at how appropriate the terms are, as we did above, is a first step. This is also known as *face validity*, as we look if our method "appears to measure" what we want it to measure (Anastasi, 1976, pp. 139–140). Yet, face validity depends on many implicit decisions that vary between context and researcher. As such, we should look further if we wish to provide a more satisfactory validation. One good candidate for this is by looking at *construct validity* (Shadish et al., 2002; Carmines and Zeller, 1979). This refers to the degree to which we can use our results to say something about that what we aim to measure. One way to learn this here is to look at the wider context in which the terms the algorithm uses appear. For example, if a term used by the algorithm to assign a speech to S occurs in a context that defines S, this strengthens our case for construct validity. To see this, we can use keyword-in-context (KWIC), which looks at the *n* (here we choose 20) words before and after the term that interests us. In Table 5 we show this for one of the terms from the PCA analysis for S – *arbetsmarknadspolitik* (labour market policy). Here, we see that the context of the word indeed refers to policies close to S. In both cases, the term is used to call for more and new measures to regulate the labour market – something indicative of S. Similar examples for the words in Tables 3 – 4 are in the online appendix. As we have implemented KWIC in our algorithm, scholars can thus easily assess whether the same is true for any of the other terms and in this way better assess the validity.

### 5.4 Explanations and Predictive Accuracy

Returning to individual instance explanations, we also wanted to investigate if the kind of words (domain specific or statistical distributions) occurring in an explanation have any relationship with the certainty of the model on those datapoints. We found domain specific words (here related to politics), along the positive PCA spectrum, while more common, general words had embeddings placing them towards the negative end. We find that data points where the explanation-words are predominantly positioned within the positive PCA

| "... enda åtgärd lösa detta, det behövs många åtgärder. Det handlar om ett gott företagarklimat, om en ny **arbetsmarknadspolitik**, om ytterligare utbildningssatsningar, om att bygga om — osv. med de förslag till åtgärder som vi ..." |
| *"... single measure solve this, many measures are needed. It's about a good business climate, about a new **labour market policy**, about further training efforts, about rebuilding – etc. with the proposed measures that we ..."* |
| "... i arbete det finns individer som kommer att behöva säskilt stöd, och då behöver vi ha en bra **arbetsmarknadspolitik**. Men det är förstås inget egenvärde i att ungdomar som kan få jobb ändå ska vara i en ..." |
| *"... in work there are individuals who will need separate support, and then we need to have a good **labour market policy**. But of course there is no intrinsic value in young people who can get a job still being in a..."* |

Table 5: Keywords-in-context for the class-explanation feature *labour market policy* for the Social Democrats.

spectrum (the sum of the PCA coordinates of the top-ten explanation features is positive) are cases where the model is more accurate. Compared to datapoints where explanations lie in the negative PCA space, there is an accuracy gain of roughly 10 percent (Table 6). Interestingly, this suggests that explanations containing domain specific, rarer words are correlated with the model's correctness, although the number of datapoints with domain specific explanations is quite small.

|              | Correct | Incorrect | Acc   |
|--------------|---------|-----------|-------|
| **Pos PCA sum** | 186     | 25        | 88.15 |
| **Neg PCA sum** | 1413    | 376       | 78.98 |

Table 6: Classifier performance on the validation set split based on the sum of PCA coordinates of the explanation provided by LIME.

## 6 Comparison to SP-LIME

Our method is comparable with SP-LIME, which aggregates individual LIME explanations. SP-LIME consists of three similar steps: post-hoc instance explanations extraction, sorting and exam-

| Rank 1 SP-LIME example (true label S): |
| --- |
| **är (is)**, *det (the)*, *som (as)*, *den (the)*, *vi (we)*, *Natomedlemskap (NATO membership)*, *att (to)*, *du (you)*, *samlingsregeringen (the coalition government)*, **Vi (We)** |
| **Rank 2 SP-LIME example (true label M):** |
| *frågorna (the questions)*, **protektionistiska (protectionist)**, **önskar (wish)**, **Det (The)**, **och (and)**, **Herr (Mr)**, *oerhört (incredibly)*, **handelsminister (Minister of Trade)**, **tackar (thanks)**, *de (the)* |
| ... |
| **Rank 12 SP-LIME example (true label M):** |
| **medelinkomsttagare (middle income earner)**, **avregleringar (deregulations)**, **vänster (left)**, **tvivelaktiga (questionable)**, *skattesänkningar (tax cuts)*, **Då (Then)**, **och (and)**, **Man (One/third person singular)**, *bostadsmarknaden (the housing market)*, *stöd (support)* |
| ... |
| **Rank 16 SP-LIME example (true label S):** |
| *borgarna (the bourgeois)*, *oss (us)*, *långtidsarbetslösa (long-term unemployed)*, *klyftorna (the cleavages)*, *det (the)*, *sjuka (sick)*, **rödgröna (red green)**[6], **Vi (We)**, **Låt (Let)**, *är (is)* |

Table 7: Explanations provided by SP-LIME. Bold features indicate words contributing towards an M classification, while italic features do the same for S. Full results are in the online appendix.

ple extraction. In contrast to our proposed scoring functions, SP-LIME calculates the score for feature $j$ as $I_j = \sqrt{\sum_{i=1}^{N} W_{ij}}$ where $N$ is the number of data points in the explained dataset and $W$ is the explanation matrix containing the local importance of the features. Based on this scoring, SP-LIME performs a greedy search to extract the top scoring data examples that also have the greatest coverage of distinct features. Therefore, the model explanation takes the form of a set number of text examples with their corresponding instance explanations, where the number of examples provided is defined by the user. Since the method performs a greedy search, the results are ordered by their contribution to how well they explain the model and how many unique features they cover.

We apply SP-LIME to the BERT classifier and extract the top 20 text examples that the explain-

ability approach considers most representative. These contain 9 S examples and 11 M examples. A selected set of instance explanations can be seen in Table 7 and the full list is available in our online appendix. We can see the overemphasis of stop words especially in the top examples. Only a couple of the surfaced terms carry a political significance, and even those lack context and have debatable generalisability. Some of the examples provided by SP-LIME (see Top 12 and Top 16 in Table 7) are instances where human intuition is easier to align with. However SP-LIME in general does not provide a way to distinguish between the two types of contributing features that the current work targets. Finally, SP-LIME also differs from our method in the way it presents texts containing explanatory features. SP-LIME tries to find texts which have as many features as possible in one and the same text, while we choose to present many alternative contexts in which explaining feature words appear, motivated by social science use-cases.

## 7 Conclusion and Discussion

We have developed a new algorithm for extracting class explanations, which takes the distinction between stop words and content words into account. It thereby provides an alternative to prior methods like SP-LIME, which mixes explanations based on e.g. stop word frequency with the presence of certain domain-specific terms. Our motivation comes from the idea of human-grounded explainability: a useful explanation for a human will focus on content rather than stop words, while still being true to the model. In our case study, we demonstrated this for speeches from the Swedish parliament, with the task of explaining a binary classifier associating speeches to either of the two main parties. This is a difficult task, our human annotation experiment showed humans performing just better than random, potentially as they primarily looked for clues about policy. The machine learning models performed better, as they likely also managed to identify statistical speech patterns of speakers, which we saw in explanations where e.g. stop words inevitably appear. Our algorithm can not only identify these, but also separate them from explanations containing domain-specific words, hinting at policy, motivated by the needs of social scientists. Additionally, we find indications that domain-specific explanations cor-

relate with model performance. Patterns related to policy in our experiment may be more robust than learned speech patterns of stop words, which risks being influenced by single frequent individuals in the dataset, rather than capturing patterns common to a political party.

Future work will focus on systematic and extensive testing of the proposed methodology in order to evaluate it along the twelve properties proposed by Nauta et al. (2022). The focus should be on measuring the faithfulness to the underlying black box model, *correctness*, as well as a larger scale domain expert evaluation to measure how relevant and valid the explanations are (*context* and *coherence* properties). The generalisability will also be tested, by studying other domains and classification tasks.

## Acknowledgments

This work was supported by the Wallenberg AI, Autonomous Systems and Software Program – Humanities and Society (WASP-HS) funded by the Marianne and Marcus Wallenberg Foundation and the Marcus and Amalia Wallenberg Foundation. RJ was supported by the Wallenberg AI, Autonomous Systems and Software Program (WASP) funded by the Knut and Alice Wallenberg Foundation.

The computations were enabled by resources provided by the National Academic Infrastructure for Supercomputing in Sweden (NAISS) partially funded by the Swedish Research Council through grant agreement no. 2022-06725.

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
