# OpenReview forum: "Class Explanations: the Role of Domain-Specific Content and Stop Words"
_NoDaLiDa/2023/Conference — NoDaLiDa 2023_

### Official Review · Reviewer_k5Cc · 2023-02-27
**Good paper on a method to exploit the distinction between function and content words when presenting users with features that explain a model,performance.**

**Rating:** 7
**Confidence:** 4

**Review:**

The paper presents a new methodology to extract features for human-centric model explanation in such a way that content and function features are treated differently and separated from each other.
There is an interesting account of existing methodologies followed by a description of the method. The method is applied to the explanation of the results obtained by BERT on  binary classification of political speeches as belonging to one of two Swedish political parties.
The paper is interesting, and the method seems novel to me.
I only have one general issue with the fact that the terms function words and stop words are used interchangeably. The concept of function word, however, is anchored in linguistic theory whereas stop words come from information retrieval and are a much looser concept. The authors should go for one of the two, and provide a definition of the way they are using the term.
On a smaller note:
l. 123, it it => it is
l. 124, independent of => independently of
l. 364, This is in line with the definition of stop words: What definition?
l. 382-386, I cannot parse these lines
l. Table 2 and 3. How do you explain the fact that “budgetpropositionen” appears in both classes?
l. 754, arguable generalisability: do you mean “dubious”?
l. 792, more easy => easier
l. 798, texts which has => texts which have


**Paper Type:**

Long paper

---

### Official Review · Reviewer_1dsa · 2023-03-09
**Good paper on explainability, minor issues with result reporting**

**Rating:** 7
**Confidence:** 3

**Review:**

The authors approach the task of neural model explainability through class explanations and demonstrate their findings on a dataset of political debates.

The authors do a good job of introducing and motivating their proposition. The paper is well-written, with a clear walkthrough of the methodology, and a substantial and thorough background overview.

There are, however, several minor issues with the reporting of experiment results. First, it would improve legibility if the scores reported in the text (Section 5) were consolidated to a table, or several. Second, judging from the preprocessing cleaning step explained in Sec. 4, should terms referring to parties, such as those shown in Table 3 (e.g. "samslingspartiet", or "kd") even be present in the dataset? If that is correct, it would be helpful to clarify the distinction between acceptable and unacceptable identifying terms in the cleanup section. The results reported in Tables 2 and 3 seem to indicate that the content of the content words is still more informative than the content/function words divide.


Typos and minor errors:

124: attempts [to] explain

384: in sufficiently distinct?

510: unemployed

796: differ[s]

798: singular or plural text?


**Paper Type:**

Long paper

---

### Decision · Program_Chairs · 2023-03-17

Accept